# Modeling for Rapid Systems Prototyping: Hospital Situational Awareness System Design †

**Avi Shaked** 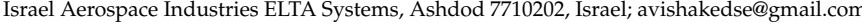

Israel Aerospace Industries ELTA Systems, Ashdod 7710202, Israel; avishakedse@gmail.com

† This paper includes a significant elaboration of the work "Shaked, A. On the Road to Hospital Digital Transformation: Using Conceptual Modeling to Express Domain Ontology". In Proceedings of the 12th International Joint Conference on Knowledge Discovery, Knowledge Engineering and Knowledge Management (IC3K 2020), Online, 2–4 November 2020.

**Abstract:** The COVID-19 pandemic caught hospitals unprepared. The need to treat patients remotely and with limited resources led hospitals to identify a gap in their operational situational awareness. During the pandemic, Israeli Aerospace Industries helped hospitals to address the gap by designing a system to support their effective operation, management and decision making. In this paper, we report on the development of a functional, working prototype of the system using model-based engineering approach and tools. Our approach relies on domain-specific modeling, incorporating metamodeling and domain-specific representations based on the problem domain's ontology. The tools practiced are those embedded into the Eclipse Modeling Framework—specifically, Ecore Tools and Sirius. While these technological tools are typically used to create dedicated, engineering-related modeling tools, in this work, we use them to create a functional system prototype. We discuss the advantages of our approach as well as the challenges with respect to the existing tools and their underlying technology. Based on the reported experience, we encourage practitioners to adopt model-based engineering as an effective way to develop systems. Furthermore, we call researchers and tool developers to improve the state-of-the-art as well as the existing implementations of pertinent tools to support model-based rapid prototyping.

**Keywords:** model-based engineering; rapid prototype development; domain-specific models; digital twin; Eclipse modeling

## 1. Introduction

### 1.1. Operational Motivation and Case Study Background

COVID-19 caught hospitals unprepared. Healthcare services—hospitals included—have been faced with the need to treat patients in isolation and remotely. In Israel, for example, hospitals opened dedicated departments for treating COVID-19 patients. Accordingly, hospitals sought effective means to manage their operations and resources and ultimately provide better service to their customers (i.e., their patients). Hospital operation can be viewed as a product–service system of the result-oriented type [1], as hospitals need to service patients with the goal of treating them as the functional result and by using appropriate means (resources and products) to do so [2]. Hospitals in Israel are faced with the need to manage their operations and resources more efficiently [3]. This has increased due to the COVID-19 pandemic, with the need to treat isolated patients with limited means. Israel Aerospace Industries (IAI) engaged with a few hospitals to utilize its technological and engineering proficiency to develop systems that may assist hospitals to operate better.

Specifically, one of the required solutions was a hospital situational awareness (SA) system. SA systems extensively rely on information to establish the perception, comprehension and status of objects and events. Such systems are considered to be a solution that addresses the aforementioned needs in support of the effective management of operations

and resources [4–6]. These systems need to structure and process information as well as to effectively communicate the information and its analysis to different stakeholders who are prospective system users [7,8], and in healthcare—as in other domains—to external stakeholders that are responsible for governance [4,9]. Information Technology (IT) systems are an enabler of modern healthcare [10,11]. In order to effectively communicate the information with various stakeholders, to produce insights with respect to the information and to support information-based collaborative work, the information should be well structured and clear, preferably standardized [12–14]. An information-driven SA system can be designed as a digital twin of a product–service system and may therefore provide the desired—but challenging to obtain—service context for healthcare product–service system development [15–17]. Furthermore, having an SA system which reflects the design of the hospital and its services as well as their up-to-date situation is well aligned with two dominant roles of product–service systems in healthcare: a design tool (e.g., a tool for designing hospital services and business) and a systems thinking decision-support tool [2].

### 1.2. System Development Concerns

SA systems are designed to communicate information in support of decision-making via representations [18,19]. The use of ontologies in designing the information embedded in such systems is essential to establishing explicit, sharable, reusable and interoperable knowledge representations [14,20–22], and it enhances context-aware capabilities in product–service systems [17]. Research efforts have produced a multitude of healthcare related ontologies, such as an ontology for healthcare technology innovation [13], ontologies describing a ubiquitous computing environment for healthcare [23,24], ontology for healthcare networks [25] and breast cancer imaging ontology [26]. While crucial for the organization of knowledge, research-derived ontologies often remain theoretical. For example, a pertinent ontology for medical services [27], which was designated to be used by IT systems, has only been checked with respect to its theoretical consistency and has not been validated for practical usability.

The not-for-profit organization Health Level Seven International (HL7) leads the "Fast Healthcare Interoperability Resources" (FHIR) specification in an attempt to standardize healthcare data from a system development practitioner perspective [28]. While FHIR includes some ontology-related concepts, these are presented from a technical implementation viewpoint and thus require significant effort to analyze and review for conceptual modeling usage; therefore, this approach was deemed inappropriate for our SA system development effort. As an illustration, in FHIR, a patient's relation to a doctor is not directly expressed; instead, it is represented by a relation between a patient and a more generic entity of the "general practitioner," which includes a relation to a "practitioner role" entity that may be assigned a specific value code to indicate that this practitioner is a doctor. This relation is directional, from the patient to the practitioner, meaning that a stakeholder who wishes to explore the ontological concepts of a doctor as a practitioner cannot identify this relation to a patient without exploring the underlying resource model from a patient perspective (i.e., the doctor and patient relation is not accessible from the doctor perspective). Furthermore, the relations are not shown as a cohesive visual representation, and this hinders the communicability of the ontological concepts.

Model-based development is an approach to engage with large amounts of data and complex information by applying appropriate formal models to domains of interest, thereby enabling rigorous, information-driven interpersonal communication, marketing research, decision analysis and impact analysis [21,29]. Practitioners state several model-based system development benefits that are pertinent to our hospital SA case, including better communication and improved system understanding [30]. Furthermore, the approach is considered an enabler for digital twin representations that can be used to validate systems with respect to real world data, provide decision support and alerts to users, predict changes in the physical system over time and discover new application opportunities and revenue streams [31,32]. All of these relate to the aforementioned operational challenge of

effectively managing the hospital operations and resources. For example, the real-world allocation of resources (e.g., doctors, monitoring devices) and of patients to departments—captured in an information model—can be used to validate the hospital model of operation, provide decision support for the acquisition of additional resources and/or placement of new patients, alert relevant stakeholders once an overcapacity condition is detected and examine how new operating paradigms can be used to improve the hospital operations and—consequently—its profits.

However, model-based development requires high expertise and is typically cumbersome to implement [32,33]. Specifically, multiple views are required to detail the composition and behavior of the developed systems, as regularly reported in modeling research [34–36], and these are difficult to communicate with stakeholders of diverse backgrounds (including nontechnical staff) [21]. A recent technical report on the maturity of the related model-based systems engineering (MBSE) approach confirms that the maturity of using models in technical processes and their management in industry is very low [30]. Specifically, the cost and time of using MBSE is primarily perceived as an obstacle ("Managing the pressure between adopting and exploiting MBSE and its benefits versus the pressure from project and program management to maintain schedule and deliver versus time and cost"; "advocating for the additional cost and time of MBSE for smaller projects"). Accordingly, companies often favor less accurate models and less disciplined trial and error tactics for prototyping. Furthermore, the model-based development of software-intensive systems is perceived as inflexible and restrictive. Using a model-based approach in rapid development iteration scenarios (considered a key technique in prototyping [37])—which involve both design and implementation—is not considered useful, and the applicability of modeling in such cases is questioned by practitioners [33].

In this paper, we describe our experience using modeling to develop a prototype for the above-mentioned hospital SA system. Our modeling approach relies on model-based engineering approaches, specifically formal modeling using a metamodel and domain-specific representations. All of these are, in fact, manifestations of conceptual modeling as applied to the hospital in order to align the technical solution with the hospital operation. First, in Section 2, we explain our development approach. Then, in Section 3, we introduce the developed prototype, showcasing the approach in practice. Finally, in Section 4, we reflect on our experience developing the system prototype using modeling tools and suggest how such tools and approaches may be widely used by practitioners for the agile development of systems. We also discuss and demonstrate how modeling tools and underlying technologies may be further improved.

## 2. Methods

Typically, the design of systems using modeling incorporates two aspects: system structural composition and system behavior. For example, in SysML—a prominent, standardized modeling language for systems engineering applications [38]—the system composition aspect is addressed by the block definition diagram and the internal block diagram, whereas system behavior is depicted using other types of diagrams, such as the activity diagram and the sequence diagram.

We approached our system design effort more sparingly, explicitly emphasizing the compositional aspect of the system via metamodeling, while allowing for the behavioral aspect to be designed vicariously via graphical design elements. Our design approach is illustrated in Figure 1.

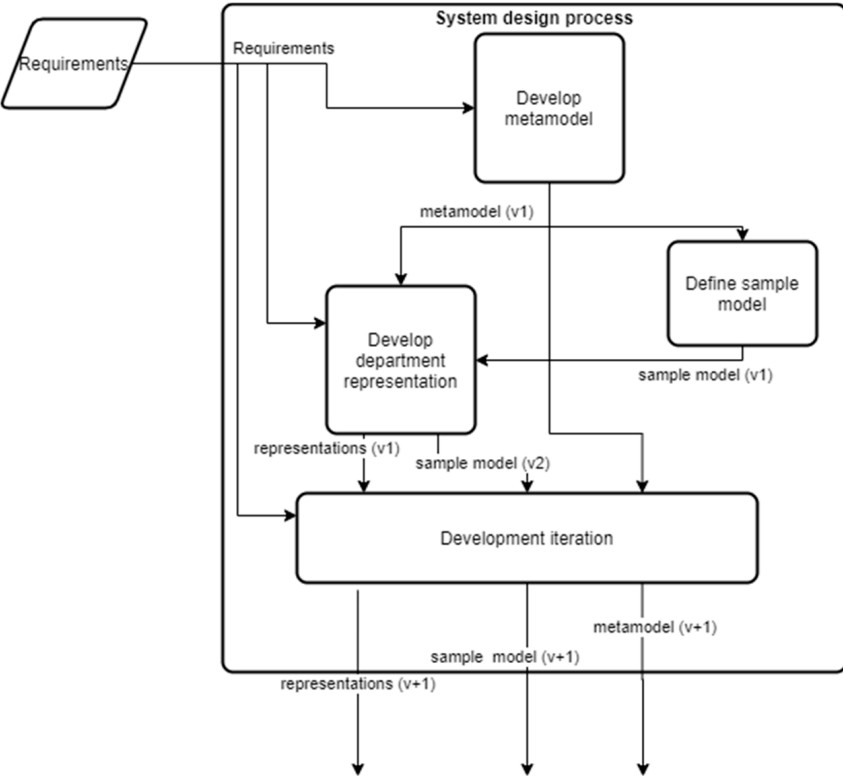

**Figure 1.** Situational awareness system design process.

Our metamodeling effort relied on the Ecore metamodel, which is used within the Eclipse Modeling Framework (EMF) for describing models [39]. The Ecore model is compliant with the Essential Meta-Object Facility specification (EMOF), which serves as a standardized and straightforward framework for mapping metamodels to implementations with the goal of allowing "simple metamodels to be defined using simple concepts" (originally in order to "lower the barrier to entry for model driven tool development and tool integration") [40]. Ecore implementations contribute to the quality of their underlying models and specifically to the technical validation of the models [41,42].

We developed an initial metamodel for the system (the "develop metamodel" activity in Figure 1). Developing this metamodel relied on a set of high-level requirements, reflecting the problem domain's ontology (i.e., concepts relating to the hospital operation). First, we derived the ontology for hospital operations primarily by examining the requirements specification and identifying the relevant ontological entities and their relationships that appeared in the specification. The requirements specification was a preliminary set of requirements for a hospital SA system. While the requirements specification is considered intellectual property—and therefore cannot be reproduced here—we address the relevant aspects. The specification was in Hebrew and comprised three sections: a mission statement, describing the objectives; a graphical figure, illustrating the operational scenario; and a list of high-level, natural language requirements, describing both medical and technical needs. While some entities and relations were mentioned explicitly in the requirements specification, others were mentioned implicitly; e.g., by a business process description. Furthermore, during the specification analysis, we identified some gaps, implying that some of the domain knowledge remained tacit (i.e., it was not stated in the requirements). Whenever deemed critical, we filled in the gaps by suggesting additional entities and relations.

The aforementioned approach was a part of an overall rapid prototyping approach taken due to the circumstances in question. Urgent hospital needs (due to the COVID-19 pandemic) and the low availability of relevant hospital personnel to provide us with feedback drove us to communicate our understanding of the pertinent domain ontology on the basis of a system prototype artifact: the formal Ecore metamodel.

Next, we defined a sample model (the "define sample model" activity in Figure 1), utilizing and manifesting the concepts defined in the metamodel by means of concrete instantiation. Once an Ecore metamodel is defined, the Eclipse integrated development environment supports the automatic generation of the metamodel as code, making it immediately available for constructing information models based on the defined metamodel.

Then, we used Sirius—another EMF component—to develop the first graphical representation of the proposed system, using graphical design elements and data queries (concerning the information captured using the underlying metamodel). Sirius is designed to allow a user to create a dedicated graphical modeling workbench [43]; however, we used Sirius in this context to create a functional application prototype. We chose to develop a representation for a hospital department first (the "Develop department representation" in Figure 1), as our analysis of the requirements established this as a minimum viable product, incorporating both the data input to the system and a situational awareness view of the data.

Figure 2 shows a representative example of a representation design specification using Sirius, revealing some of our design (which is further discussed in the following sections). Graphical elements were used to compose the representation in the Sirius Specification Editor (top section). The properties window (bottom section) allowed us to define each specific representation element, including by defining queries with respect to the underlying metamodel. As an example, the figure shows the four representations that are discussed in the next section of this paper: department patients (abbreviated henceforth as "department"), hospital map, locations and locations tree. The department representation is expanded to show its top-level representational container elements: "department" (with its lower-level container "Patient Details") and "doctorsContainer." An example of a query—starting with "aql:"—is shown in the properties window, according to which the style of the "Patient Details" container element is determined (as "gradient white to light_green" beneath the highlighted line in the Sirius specification editor window).

The designed department representation was applied to the sample model in a trial and error approach until deemed satisfactory. This approach was enabled by Sirius, which allows changes to the design of representations to be applied immediately to a given model's representation.

Repeated development iterations followed. Figure 1 depicts a single, representative iteration ("development iteration"). Each iteration was based on the previously established metamodel, representations and sample model, and yielded newer versions of these. Additionally, each iteration was developed while considering the initial set of requirements in order to keep the design aligned with the high-level objectives.

Developing a system is a creative process. The results discussed in this paper relate to a specific implementation; i.e., a specific creative process that was used to develop the system prototype. While alternative implementations would have yielded different system prototypes, the focus of surveying the specific development process—as it occurred in reality—is on demonstrating the applicability of the proposed model-based prototyping approach and communicating the state of the art and related issues in such implementations.

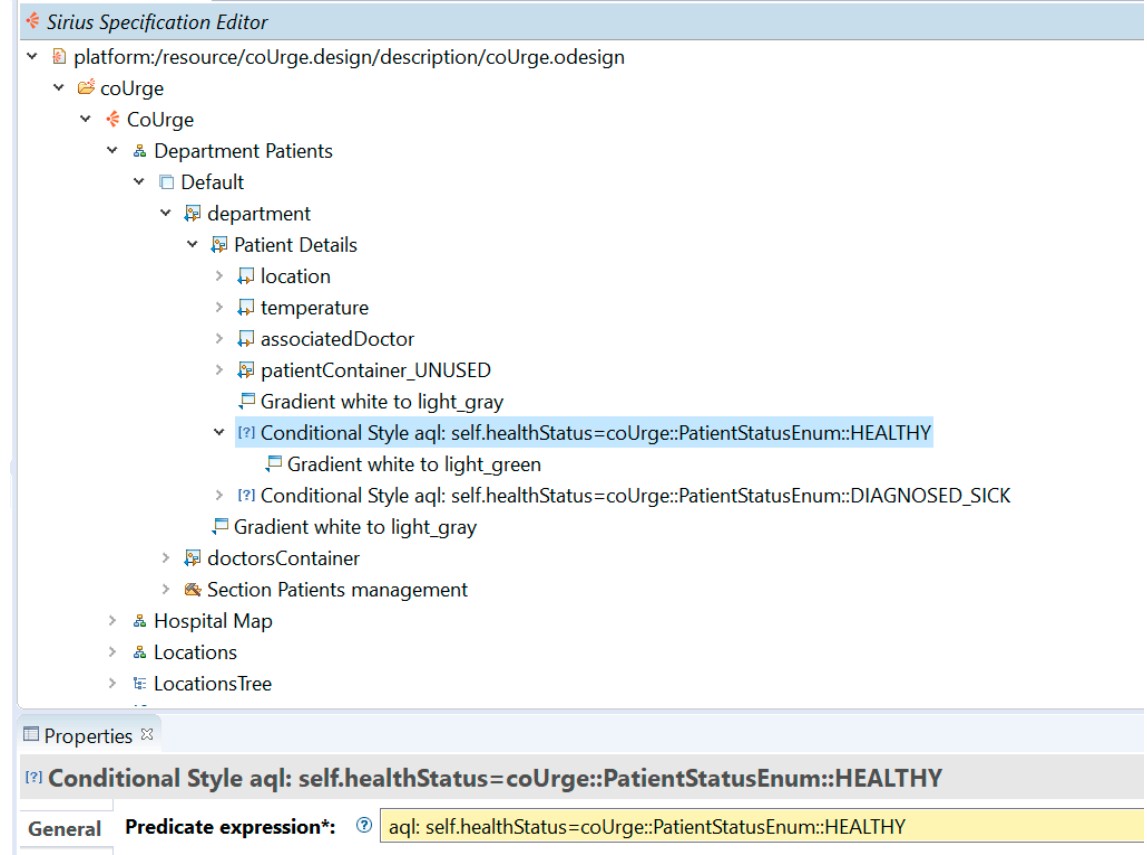

**Figure 2.** The Sirius graphical representation design interface.

## 3. Results

In this section, we present the developed hospital SA system prototype. The prototype is primarily concerned with addressing the preliminary set of requirements, by demonstrating the ability to manage hospital-related entities in an information model and support its visualization for the management of patient care within departments and of available resources. First, we introduce the metamodel which was used to organize the information within the system. Then, we disclose several representations that formed part of the system design and demonstrate how these relate to the metamodel.

### 3.1. Metamodel

The system's metamodel is shown in Figure 3. This representation shows the ontological entities as rectangular nodes (colored either yellow or grey) and their relations using edges. Relations take various forms: a directed arrow with a diamond source marks a composition relation (i.e., the source contains the target), a bi-directional arrow designates a bi-directional relation, and a hollow-headed arrow depicts a "type-of" relationship, indicating that the source entity is a type of the target entity. For example, some entities—individually—are a type of the general entity, which is used purely from a modeling perspective to add generic features (e.g., the "name" attribute, contained within the "GeneralEntity" node). The cardinality of the relations is marked as a textual tag on the opposing end of the relation edge (for example, the relations of a doctor to multiple patients are denoted as "[0..*] patient"; while the patient's singular location is denoted as "[0..1] location", with the zero (0) option indicating that no location has been assigned).

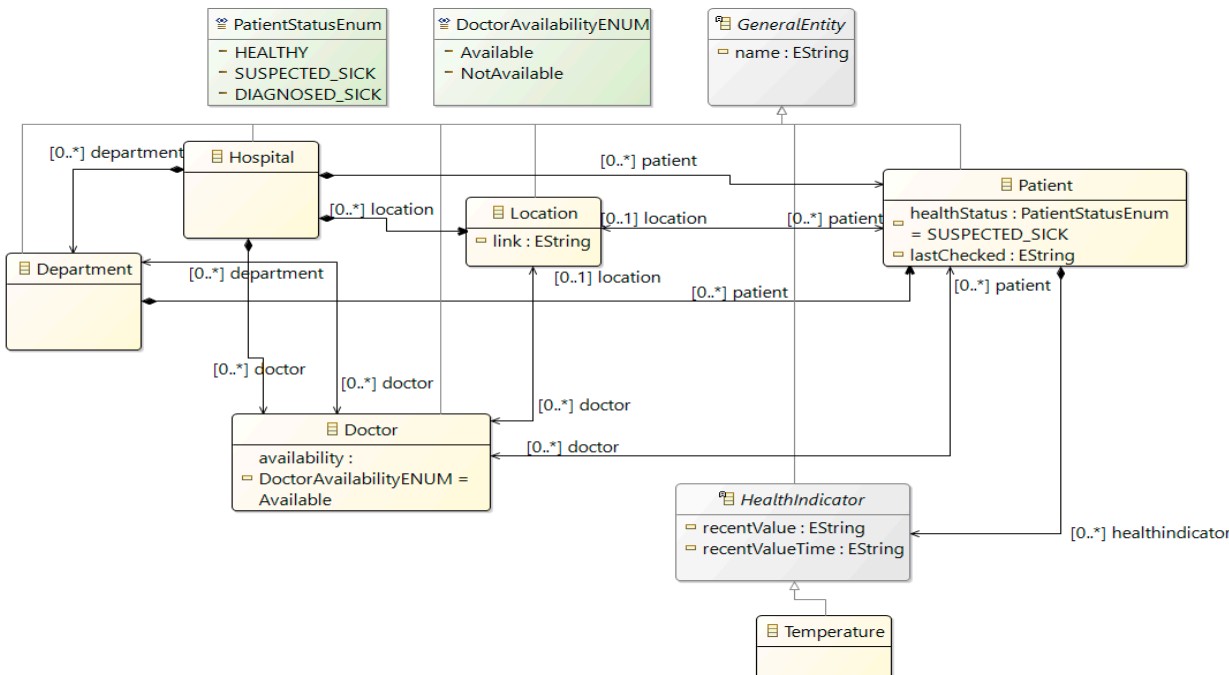

**Figure 3.** System prototype metamodel (illustrated using Ecore Tools [44]).

The metamodel reflects the ontological concepts of the hospital situational awareness domain (derived from the requirements set). A hospital entity is the top-level (root) element of the metamodel and comprises the following entities: department (conceptual entity), doctor (physical entity), location (physical entity) and patient (physical entity).

In the specific hospital scenario we were addressing (for the development of the SA system), departments, doctors and locations were all considered direct resources of the hospital. Accordingly, in our prototype implementation, the elements represented by these entities serve to illustrate the possible organizational resources, with possible relations between these elements demonstrating their possible allocation and management. Patients are organized in departments that are responsible for their treatment. Patients exhibit health indicators, such as temperature (the only health indicator represented in the version presented in Figure 3).

The relations (between entities) and their cardinalities were not as explicit in the requirements specification as the entities, and specifying many of them involved interpreting the specification. The few exceptions are as follows: (1) temperature is mentioned as a type of a health indicator; (2) location is explicitly mentioned in relation with the patient and with the doctor (however, the nature of these relations remains implicit); (3) location is explicitly mentioned as "inside the hospital," which implicitly leads to a composition relation (i.e., the hospital has locations); (4) doctors are mentioned in one statement as "belonging to the hospital," which implicitly leads to a composition relationship (i.e., the hospital has doctors); and (5) health indictors and patients are explicitly mentioned as a construct state, suggesting a composition relation between them (i.e., a patient has health indicators).

The aforementioned entities have various attributes and exhibit different characteristics, and these appear either as modeling attributes (placed within an element container), such as the "healthStatus" (read: health status) of the patient and "availability" of the doctor or as relations between the elements (marked as a textual tag with their cardinality, as previously mentioned).

### 3.2. Representations

In this subsection, we discuss three representations developed as part of the system prototype design. All of these representations rely on a specific model instantiation of the

aforementioned metamodel. This instantiation is our sample model, and is henceforth referred to as "the model."

### 3.2.1. Department Representation

This representation, which is based on the Sirius Diagram Description mechanism, is regarded as the principal representation. It is designed to depict the operational status of a hospital department (Figure 4). Specifically, it shows the hospitalized patients using a dynamic representational structure—the aforementioned patient details container—reflecting information about each department's patients. Several attributes are mentioned explicitly: name; location, based on the location resource assigned to the patient; treating doctor, based on the doctor resource assigned to the patient; and temperature, based on the reported temperature health indicator of the patient. The number in the header of the department container (value 5 in Figure 4) is dynamically updated according to the number of patient elements associated with the specific department in the underlying model. Another container in the representation shows a list of the doctors allocated to the specific department in the model.

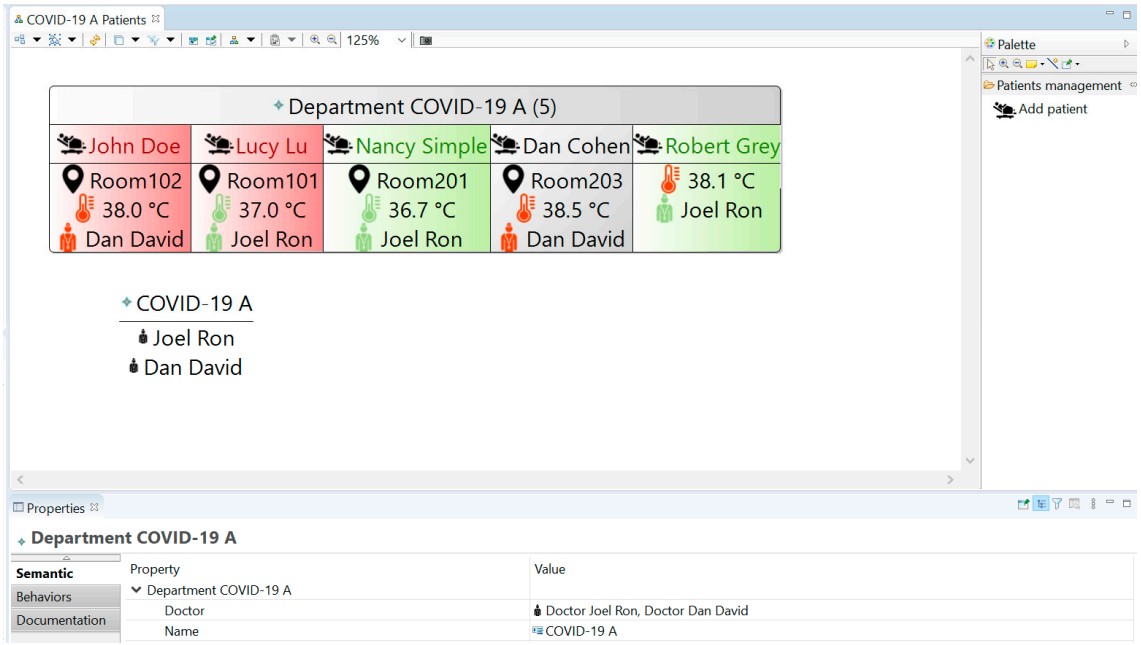

**Figure 4.** Department representation.

Additional notation is used to communicate the main aspects of a patient's treatment. Specifically, in the prototype, the coloring code of the temperature symbol indicates if the temperature is above a critical temperature (red if above 38 degree Celsius, which has been defined as an indicator of COVID-19 patient illness; green otherwise), exemplifying the ability to reflect information-based insights in real-time. Another patient attribute—healthStatus—is depicted by using a coloring scheme applied to the patient details container: grey for SUSPECTED_SICK, red for DIAGNOSED_SICK and green for HEALTHY (all values are defined as enumerations in the metamodel, as shown in Figure 3). Similarly, the availability of the treating doctor is indicated by either a red or green doctor symbol, referring to the doctor being unavailable or available, respectively.

Another aspect of the department representation is the incorporation of various tools into the representation (via the same Sirius design interface depicted in Figure 2). Tools may be in the form of an explicit visual tool, placed in a toolbox and providing a graphical user interface capability for adding and/or editing the model, or in the form of a mechanism associated with the representation of model elements, such as dragging or clicking on elements. In our prototype, the "Add patient" tool demonstrates the former and

is placed on the representation's toolbox (right section of Figure 4). Specifically, the tool in question is designed to allow the department operational staff to check in a new patient. When this tool is used, a new form opens (Figure 5). This form functions as a manifestation of the desired patient check-in operational process (e.g., entering relevant data for any hospitalized patient, such as the patient's temperature, and assigning a doctor from the available doctors). Upon completion and confirmation (by clicking the "OK" button), the tool is designed to create a new patient element under the department element in the model, set its attributes (such as the name) and create additional model elements (such as the temperature indicator, created and associated as a sub-element of the patient). Collectively, this tool and its application demonstrate a dynamic behavior of the system as it implements a specific process design (which is a part of the service design). This process design is embedded into the representational tool design (in the Sirius specification of the design, as mentioned in the previous section and shown in Figure 2). For our prototype, the temperature indication is included in the form in order to demonstrate support for a defined check-in procedure that requires the measurement of a patient's body temperature upon arrival.

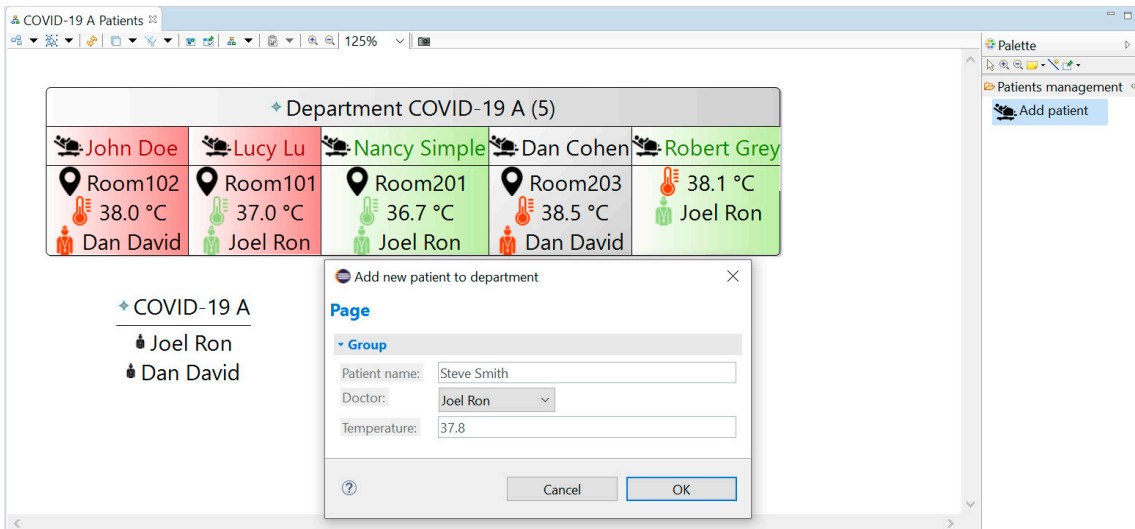

**Figure 5.** Department representation with the add new patient form.

Another tool available in the department representation is a double-click tool applied to the patient's location. The double-click tool is a built-in Sirius feature that allows a mouse double-click on a diagram object to be captured. We used this tool to initiate a dedicated function—our only extension to the Sirius modeling infrastructure—that opens a web browser with the address specified in the location's "link" model attribute. This Java function is fairly simple, featuring two instructions: it stores the location's link content (extracted from the respective model element) as a variable that holds the desired address and then opens this web address using an external web browser.

An additional double-click tool is applied to the patient details container and provides a "Patient Details" form. This form is shown in Figure 6, and while it currently resembles the "Add new patient to department" form (Figure 5), it is a different form. Using this form, a system user can only view patient details (and not change them).

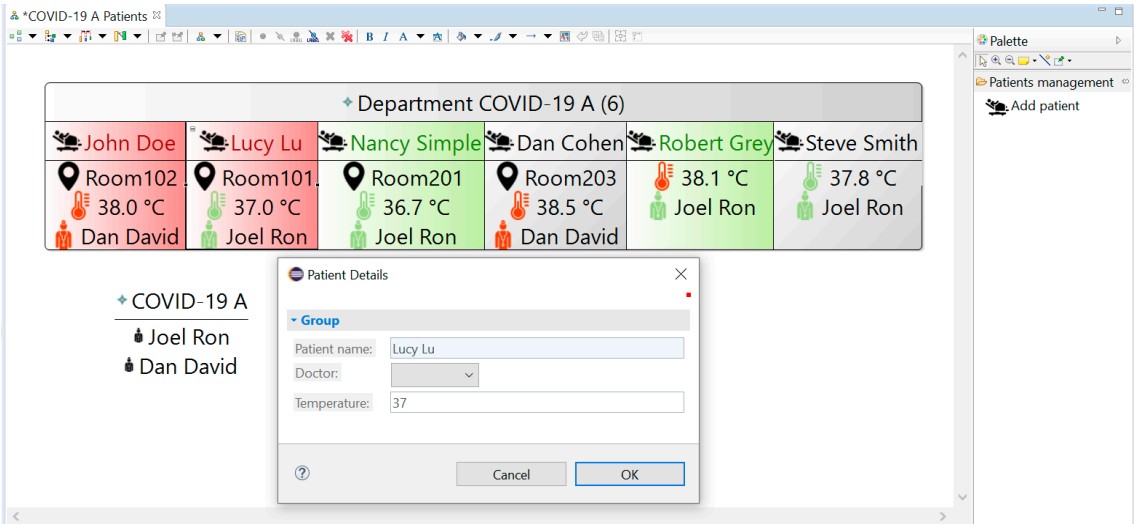

**Figure 6.** Department representation with the patient details form.

### 3.2.2. Location Representation

The location representation was developed in order to concretize the concept of location as a resource. The prototype includes two alternative designs of this representation.

The first design relies on the Sirius Tree Description mechanism and is shown in Figure 7. This design shows all of the location resources—available in the underlying model—in a hierarchical structure. This representation allows a system user to navigate the resources, whose hierarchical relations depict their physical whereabouts in the hospital. For example, in Figure 7, Room 201 is depicted as located on the second floor (Floor 2) of Building A. Furthermore, the representation places each patient element under its assigned location (e.g., Nancy Simple is located in Room 201), and this demonstrates the ability to create a hierarchical viewpoint that is different from the hierarchy used in the metamodel (we note that in our design—as shown in Figure 3—patients relate to a location by a bi-directional reference and not by a composition relation).

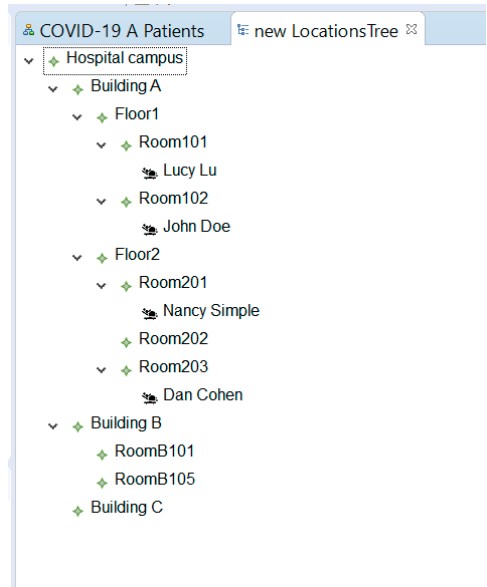

**Figure 7.** Location representation—tree design.

The second design relies on the Sirius Diagram Description mechanism and therefore visually resembles the department representation. This representation—as illustrated in Figure 8—shows the location hierarchy by using a top-level location resource as a container

for the lower level resources. A double-click tool—identical to the one previously used in the department representation—is applied to the location representational elements; and it allows an external resource—specified in the model—to be opened (e.g., the web camera feed of the location).

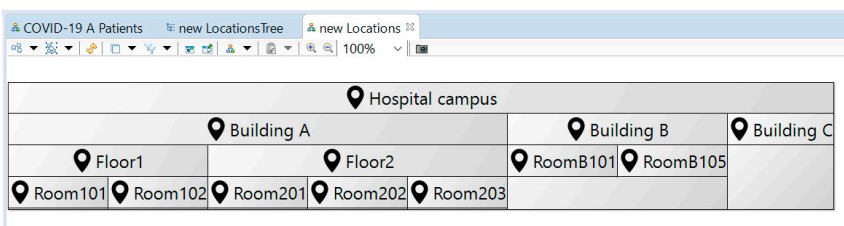

**Figure 8.** Location representation—diagram design.

3.2.3. Hospital Map Representation

The hospital map representation is another implementation of the Sirius Diagram Description mechanism that is used to exemplify a high-level, managerial view on the entire hospital's operation (Figure 9). This representation aggregates all of the hospital departments—as defined in the model—and reflects each of them using the same department container of the department representation. The comparison of the numbers at the top of each department container (presented in Section 3.2.1) can be used by hospital management to assess the workload of the departments (which was identified as an essential function of a hospital SA system [5]).

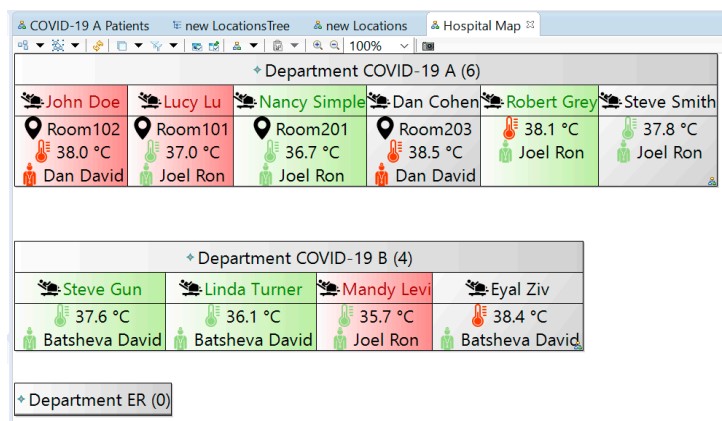

**Figure 9.** Hospital map representation.

This representation does not include the "Add patient" tool, as hospital management is not responsible for checking-in new patients. The represented information is always up to date; and changes in the departments (including new patients) are automatically reflected in the representation thanks to the model-based implementation (i.e., relying on a common, underlying information model).

## 4. Discussion

Model-based development of software-intensive systems is perceived as complex, inflexible and restrictive, often requiring users to specify multiple structural and behavioral descriptions. It is therefore not popularly used for the iterative development of prototypes. In this paper, we demonstrated that a model-based approach can be used to prototype systems by using model-based development with a twist: (a) detailing the structure for the system's information model using a formal metamodel, (b) designing system representations that capture and introduce behavioral aspects implicitly by implementing model-based tools that dynamically act on the formally structured information model,

and (c) demonstrating the prototyped system by using the representations to communicate and manipulate a sample information model.

Using a metamodel to structure the hospital SA system contributed to formalizing pertinent knowledge. Conceptual entities were clearly identified, relations between entities were concretized from somewhat implicit definitions, and their cardinality was explicitly stated, forming an ontology for hospital operations. By formally modeling the ontology, we were able to improve some relation-related definitions. While this reflects design decisions and is therefore subjective, it promotes ontology-related discussion with stakeholders, specifically with respect to the review, refinement and/or reconsideration of the system design. As our derivation of the ontology was based on specifications and not on existing ontologies, a critique may be raised claiming that our somewhat bottom-up approach can lead to an inflation of domain-specific ontologies. While existing ontologies may be used as a stepping stone for identifying a domain-specific ontology, a full investigation and/or implementation of existing ontologies can be a hurdle in practice, especially when prototyping a system. However, this should not be a barrier for ontology-based engineering [26], and introducing a grass roots ontology [45]—as we demonstrated—is a legitimate trade-off when prototyping.

The standardized implementation of the metamodel (using the EMOF-compliant Ecore) forms a basis for rigorous systems prototyping and specifically for creating functional, model-based representations. With previous research concerning Ecore-based systems implementation focusing primarily on metamodeling aspects [41,42,46], our approach offers a more holistic view of model-based systems prototyping. Our approach corresponds with a previously suggested model-based user-interface prototyping approach [47], as both rely on using a graphical language to compose representations on top of an Ecore metamodel. However, whereas the previous publication discusses prototyping only with respect to the model-based generation of a web user-interface, which remains static (post generation), our implementation is fully functional, providing users with a real model-based system experience that features dynamically updated representations and information models.

Employing a model-based design approach allowed us to explore the domain of hospital situational awareness as a hospital design challenge. Our model-based prototyping facilitates receiving timely feedback during development, which is considered highly desirable but is also extremely challenging to achieve [21]. Additionally, our approach allows for the creation of narratives, which Luokkala and Virrantaus considered essential to SA without providing a technical solution [19]. Specifically, the ability to instantiate the metamodel into concrete situations (e.g., our sample model) and demonstrate them to stakeholders using functional representations provides a possible technical solution. Characterizing the model and the representations raised multiple questions, with the intention of receiving feedback with respect to users' concepts and preferences (e.g., regarding user interfaces) as well as to the developers' understanding and suggestions. Some design-related questions were raised in the form of tangible, prototyped alternatives (e.g., the two alternative location representations). Furthermore, by incorporating tools such as the "Add patient" tool, we are able to concretize behavioral aspects of the system and of the hospital operation and communicate service design with relevant stakeholders while eliciting their take on the desirable behavior.

Our working prototype also allows us to demonstrate, suggest and explore further possibilities of using and extending the SA system. In the COVID-19 departments, patients were often placed in isolated environments and were knowingly monitored remotely using surveillance cameras. While trivial to implement, our double-click extension tool for viewing a video feed (used in several representations) was considered essential for demonstrating the ability of our SA system to interact with other systems and specifically web applications and devices. This specific demonstration allows us to show how an existing array of network-connected cameras—deployed to monitor individual patients—can be linked with the SA information system, allowing a nurse or a doctor to gain quick

access to the patient's live video feed. We are also able to demonstrate collaborative work using the system, specifically showing how changes are immediately reflected in various representations. Additionally, new features were suggested based on the location tree representation (Figure 7): the representation can be extended to accommodate patients who have not yet been assigned a location, indicating an operational problem (e.g., overcapacity, lack of resources), and the representation can also reflect the availability of unoccupied locations (and perhaps hospital beds as an additional resource), addressing the well-known operational problem of allocating rooms to patients (see, for example, [6]). The communication of our design decisions with stakeholders forms a basis not only for the system specification but also for understanding and possibly even improving operations. For example, our metamodel depicts a scenario in which the hospital manages its doctors as a common resource (expressed by compositional relation of the hospital in Figure 3) and assigns them (dynamically) to hospital departments (the bi-directional relation between "department" and "doctor" in Figure 3). This centralized approach can be contrasted with an alternative approach in which doctors are a dedicated resource of each department. The developed SA system prototype may also be further developed as a digital twin of the hospital, supporting the established roles of decision-support tools and design tools that are attributed to healthcare product–service systems [2].

In comparison with formal modeling languages—which are considered less suitable for communication with users as they are typically aimed for developers [48]—the above discussion demonstrates how our prototyping method facilitates the communication between systems designers and various stakeholders (mainly, our prospective users) using a familiar user-oriented terminology. The implementation also suggests that using a working prototype is an effective mechanism for product-service systems modeling and that it may be used for the advancement of crucial related aspects, such as business model development, enterprise transformation and the alignment between information technologies and business [45].

With respect to technical aspects, the design of our representations implicitly captures behavioral design. This is another tradeoff (in our model-based prototyping method) between being strict and being agile. Referring to the ability to reverse-engineer the prototype (for example, in order to develop a fully operational system), we consider the graphical design interface of Sirius preferable to source code. We suggest future research to examine executable graphical representations—such as Sirius—as a means to extract implicit design knowledge and perhaps even standardize a graphical notation for this purpose. We provide a manual reverse engineering example in Figure 10, demonstrating how the implicit behavior specified in the Sirius-based "Add patient" tool can be transformed into an explicit behavioral design in the form of a SysML sequence diagram. The sequence diagram is not executable by a potential user by itself and is therefore less appropriate as a system prototype (compared with our Sirius-based specification). Still, automatically generating a standardized design artifact—such as a sequence diagram—can facilitate the communication and documentation of the prototype design.

Concerning the tools used, we found both Ecore Tools and Sirius to be mature enough for prototyping. Specifically, we found the ability to immediately apply changes to the model-based representations to be very effective in prototyping. The prototype was developed by a single systems engineer in half the time it took another team of systems engineers and software engineers—working in parallel—to develop a similar prototype using a non-model-based development approach. However, since Sirius was not originally designed for prototyping systems, it lacks attractive representational elements. We suggest that additional model-based representational elements—such as gauges and indicators reflecting scales, quantitative measurements, categories, etc.—be incorporated into Sirius in order to promote its use for prototyping and, consequently, promote the adoption of model-based development by practitioners.

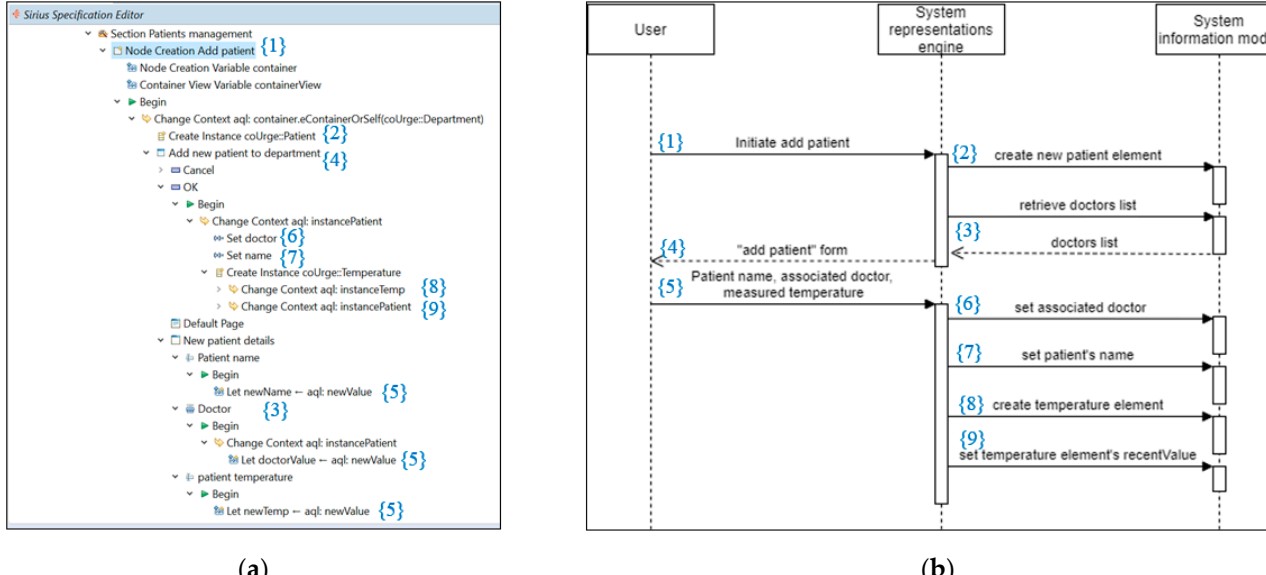

**Figure 10.** A sequence diagram can be extracted from a Sirius specification. (**a**) An excerpt from the Sirius specification, showing the graphical nodes for creating the "Add patient" tool; (**b**) a suggested sequence diagram that captures the "Add patient" design. Numbers in curly brackets are used to correlate between the Sirius specification and the derived sequence diagram.

## 5. Conclusions

In this paper, we presented and reflected on a real-world system prototype developed using a model-based prototyping approach. Our approach included the development of a metamodel—exhibiting a formal ontology derived from a set of requirements—and a set of model-based representations applied to an instantiated model. Our approach relies on existing, open-source tools to address the reported challenges of creating domain-specific models and using them productively [32]. Additionally, in the case of Sirius, we use the tool beyond its original, intended usage scenario (designing modeling workbenches) to prototype a functional SA information system. The implementation supported the demonstration of system functionality and facilitated communication with stakeholders regarding both the underlying ontology of the information system (e.g., centralized vs. decentralized resource allocation), the graphical design (e.g., feedback regarding the proposed location representation alternatives) and the operational process definitions (e.g., patient check-in). Thereby, it addresses the identified challenges of creating narratives for situational awareness [19] and of conducting a meaningful and timely discourse with nonexpert stakeholders in model-based development in order to receive feedback [21]. The implementation attests to the validity of our approach and specifically to its practical nature.

Our model-based prototyping approach alleviates some of the difficulties attributed to model-based development and, specifically, allows behavioral aspects to be captured implicitly via the rigorous, model-based graphical design of representations. Reflecting on our hospital SA system prototype implementation, we proposed further research which may improve and extend the use of model-based graphical design tools to improve the state of the art of model-based design.

**Funding:** This research received no external funding.

**Institutional Review Board Statement:** Not applicable.

**Informed Consent Statement:** Not applicable.

**Data Availability Statement:** Not applicable.

**Acknowledgments:** We wish to thank Obeo for providing us with an evaluation license of Obeo Designer Team—a commercial release of the Eclipse tools used in this work (which includes an additional, proprietary collaborative work capability).

**Conflicts of Interest:** The author declares no conflict of interest.

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
