# Peer review of "Modeling for Rapid Systems Prototyping: Hospital Situational Awareness System Design†"

_systems, doi:10.3390/systems9010012_

Round 1

Reviewer 1 Report

The paper titled "Modeling for rapid systems prototyping: hospital situational awareness system design" describes the author's experience designing a system prototype for a hospital system treating COVID-19 patients.  The author goes into extensive detail describing a meta-model without including any requirements.  The author describes the development efforts by first designing a meta-model of hospital form.  There was no rigorous description of how the model elements were instantiated.  It is not clear if other systems engineers would have designed the model similarly. 

Fundamentally, the author describes this paper as a system design paper rather than a system analysis, yet, the discussion begins with the development of system form (i.e., the meta-model) with no mention of elements of system function.  Furthermore, the author does not design the system for any explicitly stated requirements.  It is not clear why some aspects of these requirements were not included.  The description of these requirements would have made more clear how this system would be used functionally and hence how useful the developed prototype is to hospitals.  As a design methodology paper, there is no clear description of the functional domain of healthcare operations to instantiate the critical elements of form that need to be captured.  Near the end of the paper, the author models an activity-based diagram of how the user would input data into the developed prototype, but that does not address the question of how or what this prototype would be used for (patient management, clinical decision making for assigning patients to COVID floors).  A case example of how this system was used would be important to showcase this prototype. 

Finally, describing a meta-model for a system requires that a rigorous method of development of these elements be described.  It is not clear that this occurred.  There was no description of validation with any stakeholders.  The reviewer understands and concurrs with the agile method of development due to the COVID-19 pandemic, but what has been described is a system model rather than a meta-model. 

Finally, from a modeling perspective, it is not clear why the author modeled the patient as a system resource.  Instead the patient is an operand of the system.  The composition relation link between the patient and hospital in Figure 3 does not reflect the reality of the patients role in the hospital system.  

Reviewer 2 Report

Modelling for rapid systems prototyping hospital situational awareness system design

This is an interesting paper that looks into how hospital systems design can be improved by adopting an MBSE approach. The paper presents an MBSE framework (Eclipse Modelling Framework) specifically suited and adapted for hospital system design. The framework consists of dedicated tools, developed as a software prototypes.    

1) introduction

1a) I think the introduction could be shortened and focused more on the actual problem. For example, the sentence: “This hospital operation can be viewed as a product-service system of the result-oriented type (Tukker, 2004), as hospitals need to service patients with the goal of treating them as the functional result and by using appropriate means (resources and products) to do so (Xing et al., 2017)” is it needed?

1b) I would also like to read an explanation for hospital situational awareness (SA) systems. Why are they needed?

1c) I would also like the authors to elaborate clearly about the problem that motivates their approach. What is the problem with current hospital situational awareness (SA) systems, or their development? After the (rather long) introduction, I was still asking myself the question of what the problem is, and why an MBSE approach is actually needed in this context.  

2) methods

In this section, the author does a good job in describing the methods and tools used to conduct the research. However, this section shall also include details about the overall methodology adopted (i.e., data collection and data analysis).

For example, line 131: “We developed an initial metamodel for the system (the “Develop metamodel” activity in Figure 131 1). Developing this metamodel relied on a set of high-level requirements, reflecting the 132 problem-domain ontology (i.e., concepts relating to the hospital operation)”.

How did the authors collect the high-level requirements? By doing what? Reading documents? How many? which ones? By conducting interviews? How many? With who?

Also, how did the author validate the results of the model created? Was a series of interviews conducted? Or a focus group? How many?

A table and text describing more in detail the data collection and data analysis is fundamental in order to trust the results provided by the research.

3) results

Figure 4: what does the colour coding mean? (green and red? Can this be briefly reported in the figure caption?

4) discussion

4a) The discussion right now is almost five pages long, I suggest the author to summarize the section.

4b) Also, the section is a little bit mixed with the results (in my view, figure 10 and figure 11 are results).  I suggest the authors to check again the section, summarizing and moving what can be considered as results to the appropriate sections.

4c) I also suggest the author to elaborate more on the benefits of the adopted approach compared to other approaches. I suggest the author to expand some parts of the text which are very good (e.g., line 136-147) while removing some parts that are not strictly discussion and elaboration of the paper’s contribution.

4d) also, I think the author shall elaborate on the benefit of the approach in terms of the expected improvements (situation awareness and rapid prototyping). In what way the presented approach can provide these benefits? Has this been validated in some way? (e.g. a focus group, interviews).

If the approach has not been validated yet, what is the plan for future work?

5) conclusions

I suggest the author to provide a conclusion of the paper, elaborating on the main contribution of the paper.

6) Overall comment

Please elaborate here on the main elements of novelty introduced in this paper, compared to the conference paper (Shaked, 2020).

7) terminology

The authors use the term “technologies”, meaning the tools that are part of their framework (e.g. Ecore Tools and Sirius). I think this creates a bit of confusion between the tools used and the “object” of what is modelled (e.g, healthcare technologies).

I suggest the authors to replace the term “technologies” with the term “tools”. This paper could also be a useful reference for terminology (table 1): https://www.cambridge.org/core/journals/design-science/article/supporting-designers-moving-from-method-menagerie-to-method-ecosystem/63DA0F12D7C5AB2D94DDFBE40DD7E8ED

Reviewer 3 Report

In this paper the author has laid the groundwork for an interesting tool with immediate and relevant applications. A prototype of the tool, and its underlaying modeling, are presented and discussed. While model- and domain- based design approaches are not novel, the development and discussion of the prototype is valuable for future development of systems comprised of complicated ecosystems. Overall, this is an interesting paper that is well written and organized.

Round 2

Reviewer 2 Report

This is an interesting paper that looks into how hospital systems design can be improved by adopting an MBSE approach. The paper presents an MBSE framework (Eclipse Modelling Framework) specifically suited and adapted for hospital system design. The framework consists of dedicated tools, developed as a software prototype.

The author has addressed all my comments, although there is one point I would like to resists on the discussion session:

4) discussion

4a) The discussion right now is almost five pages long, I suggest the author to summarize the section.

Comment

The point is that right now the result section is 5 pages, and the discussion section is 5 pages. So, the content of result is almost as equal as the content of the discussion.

As the author correctly writes, “any comparative analysis will be purely a speculation on my side”. Therefore, I would like the paper to be a little bit less unbalanced towards discussion of topics that are not yet validated and implemented, and focus more, as the author writes “on the work that was actually done”.

For example, I now understand that “Figure 10 and 11 are not results (in comparison, Figure 3, which is the basis for Figure 10 and Figure 11, is), but rather suggestions for additional mechanisms (described in the discussion, as a reflection on the work)”.

But then, why spending three pages on discussing them?

My suggestion for the authors is to revise the discussion section and to reduce / summarize / delete the parts that are not essential to the paper, or that are left for future work. So that the discussion section can be brought to a more traditional length (approx. 1/3 of the result section).

I am sorry with the author for asking another round on revision, but I think the paper would benefit from being centred more around the results section, and not too unbalanced towards reflections and discussions (which are, by nature, a bit speculative).

I hope the author sees my point.
